# Performance and Quality Comparison of Movie Alignment Software for Cryogenic Electron Microscopy

**DOI:** 10.3390/mi14101835

**Published:** 2023-09-26

**Authors:** David Střelák, Daniel Marchán, José María Carazo, Carlos O. S. Sorzano

**Affiliations:** 1Institute of Computer Science, Masaryk University, Botanická 68a, 60200 Brno, Czech Republic; 373911@mail.muni.cz; 2Spanish National Centre for Biotechnology, Spanish National Research Council, Calle Darwin, 3, 28049 Madrid, Spain; da.marchan@cnb.csic.es

**Keywords:** Cryo-EM, movie alignment, performance, FlexAlign, MotionCor2, Relion MotionCor, CryoSPARC, Warp

## Abstract

Cryogenic electron microscopy (Cryo-EM) has been established as one of the key players in structural biology. It can reconstruct a 3D model of a sample at a near-atomic resolution. With the increasing number of facilities, faster microscopes, and new imaging techniques, there is a growing demand for algorithms and programs able to process the so-called movie data produced by the microscopes in real time while preserving a high resolution and maximal information. In this article, we conduct a comparative analysis of the quality and performance of the most commonly used software for movie alignment. More precisely, we compare the most recent versions of FlexAlign (Xmipp v3.23.03), MotionCor2 (v1.6.4), Relion MotionCor (v4.0-beta), Warp (v1.0.9), and CryoSPARC (v4.0.3). We tested the quality of the alignment using generated phantom data, as well as real datasets, comparing the alignment precision, power spectra density, and performance scaling of each program.

## 1. Introduction

The first step in the Cryo-EM processing pipeline is typically referred to as movie alignment. A movie consists of a sequence of frames produced by the microscope, with each frame recording projections of tens to hundreds of particles. By averaging these frames, a micrograph is generated, which is subsequently used for particle picking, contrast transfer function (CTF) estimation, and other stages of the image processing pipeline. However, due to factors like beam-induced motion and other alterations within the recorded area during imaging, simple frame averaging of frames is insufficient.

The causes of these variations may differ from one sample to another, and they are comprehensively described in [1]. The motion can manifest as both global, affecting the entire rigid frame, and local, affecting specific areas of the frame, and it is necessary to correct both types.

The alignment process is challenging due to the extremely low signal-to-noise ratio (SNR), as the electron arrival is expected to occur following a Poisson distribution; this means that the most common observations are 0 or 1 electron per pixel.

Typically, we differentiate between two types of alignment. Global alignment aims to compensate for the apparent movement of the entire frame. While this can result in incorrect alignment in specific areas, it enhances the overall SNR. Consequently, it is often employed as the initial step before local alignment.

Local alignment, in contrast, seeks to correct more complex particle movements caused by factors such as the beam, doming, or another cause. Typically, it operates on a divide-and-conquer basis: the movie is divided into small patches, and alignment is independently solved for each patch. Ultimately, both global and local movements are corrected by interpolating and summing values from all movie frames, thus producing a micrograph.

Three factors drive the requirement for fast and precise algorithms for movie alignment. The first one is speed. The new generation of detectors [2] and acquisition practices increased the throughput to 300–400 movies per hour [3], with plans to grow to about 1000 movies per hour. It is crucial to be able to process these movies in real time, as potential problems with the imaging or the sample can be discovered and possibly corrected as soon as they appear during the acquisition (the access cost to the microscope ranges from USD 1000 to USD 3000 per day).

The second factor is accuracy. The goal of Cryo-EM is to produce near-atomic models of the macromolecules under study [4]. The last step of the processing pipeline, 3D reconstruction, has been a hot topic for the past few years with several important studies [5,6]. However, to reach high-resolution models, all the steps in the processing pipeline have to be looked at carefully. This goal sets an important challenge to all the image processing steps, especially this one, as the SNR of the micrographs ranges from 1/10 to 1/100 (at the level of the frame, this SNR has to be divided by the number of frames, typically between 10 and 100).

The third factor is particle tracking for polishing. Being able to track the particles back to the originating frames accurately is crucial during the polishing phase, which aims to further improve the resolution of the final 3D reconstruction.

In this article, we compare the most commonly used programs for movie alignment—more precisely, we compare the most recent versions of CryoSPARC (v4.0.3) [7], FlexAlign (Xmipp v3.23.03) [8], Relion MotionCor [9] (v4.0-beta), MotionCor2 (v1.6.4) [10], and Warp [11] (v1.0.9). We tested the quality of the alignment using generated phantom data and real datasets using various metrics and performance scaling of each program on a single fat node. Typical facility installations use a fat node or a cluster installation with a naive work distribution.

The rest of this paper is organized as follows: Section 2 provides additional details on each movie alignment program. In Section 3, we describe the methodology used. Quality and performance evaluations are conducted in Section 4. We discuss the results in Section 5. Finally, conclusions and future work are presented in Section 6.

## 2. Comparison of Movie Alignment Programs

CryoSPARC allows for the selection between several implementations of movie alignment, including the MotionCor2 wrapper, patch, full-frame, and local motion correction [7]. The method used in this paper, as recommended by the manufacturer, is the patch motion method. This autotuning patch-based method performs full-frame stage drift and local anisotropic motion correction. It is accelerated on a GPU but is not open source.

MotionCor2 is probably the most commonly used software for movie alignment [10]. While MotionCor2 provides good performance and precision, to the best of our knowledge, it does not provide the data needed for particle tracking. It allows for both global and local alignment and is accelerated on a GPU. It uses cross-correlation (CC) to align frames or patches of the movie. It is not open source.

FlexAlign performs global and elastic local registration of the movie frames using CC and B-spline interpolation for high precision [8]. It is accelerated on a GPU and is open source. Interpolation coefficients are stored to allow for particle polishing.

Relion MotionCor implements Bayesian polishing (also open source) [9]. Internally, it uses a CPU implementation of the MotionCor2 algorithm. Relion does not have GPU-accelerated movie alignment, and the CPU code by default uses double precision, which further penalizes its performance.

Warp also uses CC and is GPU-accelerated [11]. Alignment information, including local alignment, is stored as an *xml* file.

A brief overview of the compared programs can be found in Table 1.

As a final note, it is important to highlight that not all movie alignment programs produce the same type of output. For instance, FlexAlign produces the average of the input frames, while MotionCor2 produces the sum of the input frames. Additionally, the interpolation scheme used to generate the aligned frames may also influence the specific values and power spectra of the output micrographs.

## 3. Methodology

### 3.1. Quality—Phantom Movies

As we do not know the ground truth for real data, we opted for using generated phantom movies, whose properties we fully know. We used the latest version of the *xmipp_phantom_movie* program available from the Xmipp suite in version 3.23.03-Kratos. Xmipp is a suite of image processing programs, primarily aimed at single-particle 3D electron microscopy, designed and managed by the Biocomputing Unit located in Madrid, Spain [12,13].

The *Xmipp_phantom_movie* program can generate movies with a specified resolution size and number of frames. The signal in the frames is represented by either an equally spaced grid, a disc, or a cross, which is not typical in cryoEM but allows us to easily study the quality of the alignment. The signal can be translated in each frame using a fixed-step shift or a more complex function. Additionally, barrel deformation can be applied to simulate doming (its formal description is given below).

Depending on the settings, the program can simulate the ice via range-adjusted, low-pass filtered noise with a normal distribution and the dose using a Poisson distribution on a per-pixel basis. To ensure the repeatability of the output, a fixed seed for random generators can be used. Lastly, the program can generate dark and gain images of an appropriate size, with values of 0 for the dark image and 1 for the gain image, which can be used for performance testing. The resulting movies and images can be stored in various formats. For this study, we opted for the *mrc* format due to its wide support among the tested programs.

To test various properties of the movie alignment programs, we generated several types of movies with various numbers of pixels:3838 × 3710 (K2 detector);4096 × 4096 (Falcon detector);5760 × 4092 (K3 detector);7676 × 7420 (K2 detector, super-resolution);11,520 × 8184 (K3 detector, super-resolution).

For each size, we have generated noise and noiseless movies with 6 and 10 frames, as typical for cryogenic electron tomography, and with 70 frames, as typical for single-particle analysis. Please note that the term *typical* should be understood in a very vague manner, as the number of frames greatly depends on the experiment setup. These movies assume that the sampling rate is 1 Å per pixel. They contain a grid with a spacing of 150 pixels, and each line is 5 pixels wide. Also, we generated noisy and noiseless movies with a size of 4096 × 4096 × 70 frames and a grid step of 300 pixels. On each frame, a fixed shift was applied so that the total shift of the movie was from 50 to 120 pixels, with a step of 10 pixels between movies. The shift has been applied to each pixel [x,y] and frame *t* using the following formula:(1)$$\begin{array}{cc} x(t) & =a_{1}t+a_{2}t^{2}+\frac{cos(t)}{10}\\ y(t) & =b_{1}t+b_{2}t^{2}+\frac{sin(t^{2})}{5} \end{array}$$
$a_{1}$, $a_{2}$, $b_{1}$, and $b_{2}$ are coefficients used to model shifts in two directions. To determine their default values, we fine-tune them until we obtain the most suitable coefficients for simulating shifts commonly encountered in cryoEM. To capture basic linear shifts, we focus on $a_{1}$ and $b_{1}$, representing the linear components of the shift. These coefficients dictate the slope or rate of change in the shift along the x and y directions. To capture more complex shift behaviors, such as parabolic patterns, we introduce coefficients $a_{2}$ and $b_{2}$. These coefficients determine the steepness and curvature of the function, allowing us to represent more intricate and non-linear shifts. We used values of $a_{1}=-0.039$, $a_{2}=0.002$, $b_{1}=-0.02$, and $b_{2}=0.002$.

Furthermore, the barrel/pincushion transformation has been applied to each shifted frame to simulate doming. For normalized coordinates ([−1,1]), the pincushion transformation changes the radial location of a given pixel as follows:(2)$$r_{out}=r_{in}\left(1+k_{1}r_{in}^{2}+k_{2}r_{in}^{4}\right)$$
where $r_{in}$ and $r_{out}$ represent the radius of a given pixel in polar coordinates of the input (before the barrel deformation) and output (after the barrel deformation) images. The *k* values were linearly interpolated between frames, from $k_{1_{start}}=k_{2_{start}}=0.01$ to $k_{1_{end}}=k_{2_{end}}=0.015$, to make the transformation more prominent in the later frames.

The frame content consists of a phantom signal—either a regular grid, a disc, or a cross, optionally embedded in ice. The Poisson arrival of electrons can also be simulated as an optional step. The process starts by creating the ice if included. The ice is simulated by a random Gaussian field (in our simulation, we used a N(0,1) distribution). Once the Gaussian field is simulated, it is low-pass filtered (up to 3Å) and re-scaled between a minimum and maximum value (0 and 2, in our case). Subsequently, the signal is added to the frame.

Next, all frames are generated by applying the deformation patterns described above. These simulated frames are deterministic, as the ice structure is simulated only once and it is deformed along the process. At this stage, we may also simulate the Poisson arrival counting. The values of the deterministic frames described above are treated as mean values of a Poisson and sampled from these Poisson distributions. The entire process is visually represented in Figure 1.

Each of these noiseless and noisy movies, with non-uniform and fixed-step shifts per frame, was processed by all programs using either default settings or the settings recommended in the documentation.

### 3.2. Quality—EMPIAR Movies

In our study, we also incorporated experimental data from three different datasets obtained from the Electron Microscopy Public Image Archive (EMPIAR): EMPIAR 10,288 [14], 10,314 [15], and 10,196 [16].

EMPIAR 10,288 dataset consists of movies with 40 frames, whose size is 3838 × 3710 pixels. The pixel size is 0.86 Å and the average exposure was 1.25 e/Å^2^. Data were acquired using a Gatan K2 SUMMIT camera on FEI Titan Krios. Gain correction images are provided.

EMPIAR 10,314 dataset includes movies with 33 frames, each having a size of 4096 × 4096 pixels. The pixel size is 1.12 Å with an average exposure of 1.51 e/Å^2^. Data were acquired using the Falcon 3EC camera on Titan Krios. This dataset does not include gain or dark correction images.

The EMPIAR 10,196 dataset contains movies with 40 frames, each having a size of 7420 × 7676 pixels and an average exposure of 1.264 e/Å^2^. The pixel size is 0.745 Å, which corresponds to a super-resolution setting. Data acquisition was carried out using a K2 camera on Talos Arctica. Gain correction data are provided, along with specific instructions for their application (rotation by 90 degrees left and horizontal flip).

The diversity in dataset characteristics and camera models enriched our analysis, offering valuable insights into the algorithms’ robustness and adaptability to different experimental setups.

### 3.3. Quality—Metrics

For both the phantom data and the real data, we collected multiple metrics for the resulting micrographs. We collected three windows of 512 × 512 pixels each: one from the top left corner, one from the center center, and one from the bottom right corner of each micrograph. Additionally, we generated a normalized center window of 512 × 512 pixels, along with a histogram of the entire micrograph before normalization. Lastly, we generated the power spectra density (PSD) plots for each micrograph.

In the case of the real data, we also collected the maximal frequency using two different methods: *xmipp_ctf_estimate_from_micrograph* and *Gctf*, both using the default parameters as proposed by respective protocols in Scipion [17].

The decision to use two different methods for estimating the limit resolution of the CTF in real data analysis was made to ensure a more robust comparison. By employing Xmipp and Gctf methods, we obtain two different ways of measuring the resolution at which the micrographs contain recoverable information. The Xmipp method estimates this value by determining the resolution at which the experimental power spectrum density (PSD) drops below 1/100 of the PSD at the origin. In contrast, the Gctf method measures the frequency at which the theoretical PSD function has a positive correlation with the PSD of the image. By using these two different approaches, we can verify that the estimated limit resolution is consistent across methods, enhancing the robustness of our comparison.

The ultimate test for real data would be to execute the entire processing pipeline till we achieve the structure’s 3D model. However, such a test is not recommended in this context. Each movie alignment program uses a different reference frame for alignment. In other words, the particle’s position differs between micrographs generated by different programs. While we could re-center each particle and keep the rest of the processing pipeline intact, we cannot guarantee that the change in the final resolution of the 3D reconstruction comes only from the substitution of the movie alignment program and not from the re-centering step and other non-deterministic computations along the way.

### 3.4. Performance

To evaluate the performance of each program, we conducted multiple executions using the same phantom input. This scenario represents the best-case situation where the data are readily available (unlike in a streaming environment where data are continuously delivered) and may even be cached in RAM. It is worth noting that the processing time for movies with a high resolution and many frames can be limited by the read speed of the storage on the processing node (for instance, a movie with a size of 11,520 × 8,184 pixels and 70 frames in the MRC format consumes 24.58 GB of storage).

For each program, we ran multiple executions on the same input, processing the movie and providing gain and dark correction images (except for Relion MotionCor and CryoSPARC patch motion correction, which do not support dark correction images). Additionally, we tested scaling using multiple GPUs and different numbers of threads (in the case of Relion). We also assessed the performance of batch processing in MotionCor2. Lastly, an additional experiment was conducted involving placing the data on a solid-state drive (SSD) for faster reading and writing operations.

As FlexAlign uses an autotuning technique to optimize its performance for specific combinations of the size of the movie, and settings and used GPU, we also measured the time of this step. The result of autotuning is stored in a file and reused when possible.

Notice that Relion MotionCor is not GPU-accelerated. By default (as installed by Scipion), it uses double precision for CPU computation.

It is important to note that direct access to the CryoSPARC method is unavailable. Therefore, we had to use an alternative approach. We called the main program, created a project, imported the movies, and connected them to the local movie alignment job. However, the performance times were measured directly from the log file provided by CryoSPARC, specifically by tracking the time it took to complete the alignment job. This ensured that the performance measurements accurately reflected the time required for the alignment process in CryoSPARC.

We executed all programs intending to correct the local alignment, and we used the default or the recommended settings for each program. FlexAlign and CryoSparc set the number of patches automatically. Still, for MotionCor2 and Relion MotionCor we used recommended values (5 × 5 for movies smaller than 5000 pixels, 7 × 5 for bigger ones) and also multiples of 2 and 3, respectively, (i.e., 10 × 10, 15 × 15×, 14 × 10, 21 × 15) to see the quality and performance difference.

We did not record the execution time for experimental movies from the EMPIAR, as the processing time should be primarily dictated by the number of frames and their size, rather than the content of the frames.

All scripts generated for these data and the movies are available on Zenodo [18] and Empiar [19].

## 4. Results

The results section of our study focuses on evaluating the performance of various alignment programs in terms of quality and time efficiency. The tests were performed on a CentOS 7 Linux server with 40 cores (2 × Intel Xeon Gold 6230, 2.20 GHz) and 384 GB of RAM. The workstation also featured four GPUs (Tesla T4 Driver Version 460.27.04, CUDA Version: 11.2) with 16 GB each. In terms of storage, it has four 8 TB SATA HDDs in a RAID 5 configuration for mass storage (where data were stored), two 1 TB SATA SSDs in a RAID 0 configuration for scratch, and two 240 GB SATA SSDs. This machine is housed within the Biocomputing Unit data center at the Spanish National Centre for Biotechnology (CNB-CSIC).

For Warp, we used a different machine with Windows 10 Pro 64-bit (10.0, Build 19045). The desktop has eight cores (AMD Ryzen 7 1700, 3.0 GHz) and 64 GB of RAM (4 × 16 GB). The single RTX 2080 Ti with 11 GB of memory uses driver 536.23. In terms of storage, it has one 500 GB HDD and one 120 GB SSD connected via SATA 600, where the data were copied before the test. This machine is housed within the Sitola laboratory, a joint facility of the Faculty of Informatics and Institute of Computer Science at Masaryk University and the CESNET association at Masaryk University in Brno.

### 4.1. Quality

The quality assessment of the alignment algorithms involved the analysis of both simulated and real datasets. For the simulated dataset, we evaluated alignment quality by examining the aspect of the signal pattern in the resulting micrographs. This allowed us to assess how well the algorithms handled deformations, noise, and shifts in the simulated data.

In the case of the real dataset, our focus was on measuring the limit resolution criteria of the CTF estimation using two different programs. This criterion provides information about the maximum level of detail that can be resolved in the images. Additionally, we examined the energy decay in the power spectrum density for higher frequencies. This analysis was conducted on three different datasets, each comprising 30 movies. By evaluating these parameters, the study aimed to gain insights into the accuracy and effectiveness of the alignment algorithms in capturing fine details and preserving image quality.

#### 4.1.1. Phantom Movies

Figure 2 presents examples of simulated aligned phantom movies. We used grid-based phantom movies to examine the grid pattern of the resulting micrographs and evaluate how well the algorithms handle common issues such as noise, deformation, and shifts. For this kind of movie, we have both noiseless and noisy versions, with the noisy ones simulating the ice and dose found in typical cryoEM experiments. Additionally, we included a cryo-EM-based simulated movie that allows for a comparison in a more realistic cryoEM scenario. These types of phantom movies were designed to be a middle ground between particle projections and a regular grid. Furthermore, all types of movies were subject to barrel deformation, which was applied to the movie frames to simulate the common dome effect observed in cryoEM. This results in a more rigid pattern in the center region, gradually curving towards the edges.

By studying the alignment quality of these simulated movies, we aimed to gain valuable insights into the algorithms’ results and their ability to accurately align movies with different characteristics and complexities. However, we cannot reject the possibility that the relative alignment accuracy of the methods may be affected by this particular type of signal.

The visual analysis presented in Figure 3 provides insights into the quality of the alignment algorithms by comparing the alignment results for pristine movies and their noisy counterparts. This allows for the evaluation of how each algorithm handles deformations, shifts, and noise present in the phantom movies.

In our experiments, we conducted tests on movies of different sizes and observed interesting trends in alignment quality. It became evident that all programs successfully aligned the center region of the movies. However, as we moved towards the edges, the algorithms encountered difficulties and exhibited poor alignments, resulting in a blurring effect in the mesh pattern. This blurring effect was also noticeable in the noisy movies, where incorrect alignment caused the mesh pattern at the edges to appear blurred. This behavior remained consistent across movies of varying sizes and frame counts. When comparing the results of different algorithms, FlexAlign, in particular, excelled at achieving a cleaner pattern in both pristine and noisy images at the edges, indicating superior alignment capabilities.

Furthermore, as we experimented with higher-dimensional images, we simulated a proportional increase in beam-induced movement (BIM) and drift. Consequently, most of the algorithms struggled to align the edges correctly and, in some cases, even struggled to align the central region in the largest dimension. These findings underscore the challenges alignment algorithms encounter when handling larger image deformations, particularly in terms of preserving alignment accuracy at the edges.

The analysis in Figure 4 allows us to observe the performance of the alignment algorithms when subjected to a 60-pixel shift. By comparing the aligned movies to the pristine and noisy versions, we can assess how effectively each algorithm handles shifts in the phantom movie.

For the experiments, we used movies of the same size (4096 × 4096 × 70) but introduced varying global drift. Specifically, we introduced eight shifts ranging from 50 to 120 pixels, with increments of 10 pixels. Similar to the previous experiment, we observed a consistent pattern where all programs successfully aligned the center region of the movies. However, as we moved toward the edges, the algorithms encountered difficulties, resulting in poor alignment.

It is worth noting that as we increased the drift most of the algorithms struggled to align the edges correctly. FlexAlign, in particular, was sensitive to these changes, as it typically expects frame movement within normal cryoEM conditions. It exhibited alignment failures at shifts over 90 pixels using the default settings. To determine whether FlexAlign could handle larger shifts, we increased the maximum expected shift parameter in the algorithm, leading to improved alignment accuracy.

These findings highlight the challenges alignment algorithms face when confronted with increasing drift levels. They also underscore the importance of parameter optimization and understanding the specific limitations and sensitivities of each algorithm to achieve accurate alignment results, particularly in the presence of significant shifts.

Figure 5 enables a comparison in a more realistic cryoEM scenario. These types of phantom movies were intended to replicate not only cryoEM conditions, such as shifts, dose, noise, and deformations but also to simulate their main signal, which is particle projections. This way, the relative alignment accuracy of the methods is not affected by the particularity of the grid-type signal studied before.

These results corroborated our earlier findings. Despite differences in signal complexity, all programs effectively aligned the central region of the movies. However, as they approached the edges, the algorithms encountered challenges and produced suboptimal alignments, resulting in visible blurring in the particle projections at the micrograph’s periphery. When comparing the outcomes of different algorithms, FlexAlign once again demonstrated its ability to excel by consistently achieving a cleaner pattern, indicating superior alignment capabilities at the edges.

#### 4.1.2. EMPIAR Movies

Table 2 compares the CTF resolution limit, expressed in Angstroms (Å), for each EMPIAR entry using the Gctf and Xmipp methods. The mean value indicates the average CTF resolution limit obtained from each respective method, while the standard deviation represents the variation or dispersion of the CTF criteria around this mean value. By comparing the mean and standard deviation values between the Gctf and Xmipp methods for each EMPIAR entry, we can gain insights into the consistency and accuracy of the CTF estimation provided by these methods.

For consistency and to assess the potentially significant differences in quality performance among different algorithms, a comprehensive statistical study was conducted. This study aimed to evaluate the significant differences in means both collectively using an ANOVA test and individually by comparing all possible program combinations through paired t-tests.

The ANOVA (Analysis of Variance) test was employed to analyze whether there is a statistically significant difference among the means of the various programs. This test allows us to determine if there are significant variations between the groups as a whole and provides an overall assessment of the statistical significance of the observed differences.

If the ANOVA test yielded a significant result, a post hoc analysis was performed to compare the means of each pair of programs individually. This approach enables us to assess the significance of the differences between specific pairs of programs and identify which programs exhibit statistically different performances.

By conducting the ANOVA test and its subsequent post hoc analysis, we can thoroughly investigate the significant differences in means between the programs under consideration. This statistical analysis enhances our understanding of the variations in quality performance among the different movie alignment algorithms in cryoEM.

Figure 6 presents an analysis of the resolution estimation based on the CTF criteria for various EMPIAR entries, facilitating a clear comparison of the resolution performance among different algorithms.

For EMPIAR entry 10,196, the ANOVA test conducted on the group means did not reveal any significant differences.

For EMPIAR entry 10,288, the ANOVA test detected a significant difference in the group means obtained from the Xmipp program at a confidence level of 0.05. Additionally, the post hoc analysis indicated a significant difference between the movie alignment program Warp and the other programs, signifying a poorer quality performance by Warp in this test. Furthermore, CryoSPARC and MotionCor2 exhibited a statistically significant better resolution limit than Relion MotionCor. Regarding the Gctf metric, the ANOVA test did not identify any significant differences.

For EMPIAR entry 10,314, the ANOVA test found a significant difference only for the Xmipp criteria, indicating that different algorithms performed significantly differently for this dataset. The post hoc analysis further revealed a significant difference between all programs and Relion MotionCor and Warp, with the latter two generating micrographs with lower resolutions than the others.

Figure 7 visually represents the observed PSD trends for each program and EMPIAR entry, enabling an analysis of how different alignments affect the PSD pattern. These plots illustrate the distribution of energy across various frequencies. Ideally, with correct alignment, different algorithms should not alter the PSD patterns. The occurrence of such alterations suggests that the algorithm introduces bias to the image.

Regarding the attenuation of PSD energy at higher frequencies, we observed varying degrees of damping among the programs. CryoSPARC and MotionCor2 exhibited minimal damping as they reached higher frequencies. Conversely, FlexAlign and Warp showed slight damping at higher frequencies. It is important to note that this damping occurs close to the Nyquist frequency, at which point the signal’s energy has largely dissipated. Therefore, this energy reduction has a minimal impact on noise reduction. Additionally, it is worth mentioning that for dataset 10,196 Warp exhibited an unusual behavior, displaying different and less favorable energy decay compared to the others. We believe this issue may be associated with Warp’s difficulty in processing this particular dataset.

Finally, Relion MotionCor appeared to lose energy across frequencies. These differences can be attributed to the interpolation function used to generate the output micrograph. CryoSparc and Motioncor2 likely use Fourier cropping, FlexAlign and Warp employ B-spline interpolation, and Relion uses linear interpolation. Relion’s damping begins at lower frequencies than those detected by XMIPP and Gctf. This observation suggests that we may have had sufficient energy to accurately estimate the CTF and extract the resolution limit at higher frequencies. Furthermore, the XMIPP criteria appear to be more consistent with the damping value since its resolution limit is based on the experimental PSD decay rather than the correlation of the theoretical PSD function with the experimental PSD.

To observe how the alignment by various algorithms impacts image content in real space, Figure 8 visually illustrates the distribution of pixel values in the aligned movies generated by various algorithms, providing a means for comparative analysis of pixel value distribution across different alignment programs. Our experiments revealed variations in the pixel values of the resulting aligned images, depending on the specific algorithm used.

To assess whether there were statistically significant differences in pixel value distribution, we normalized images produced by different alignment algorithms and conducted pairwise Kolmogorov–Smirnov tests to compare them. This non-parametric test evaluates whether two datasets share the same underlying distribution. In most cases, the test rejected the null hypothesis, indicating that the two datasets originated from distinct distributions. This highlights that achieving consistent information content from the alignment of the same image with different algorithms is not as straightforward as image normalization. Instead, it underscores that different alignment algorithms can yield statistically distinct pixel value distributions.

The coefficients of variation (CV) reveal the extent of variability around the mean of all pixel values, even when considering outliers. Across all distributions, we observed low to moderate CV values, indicating that pixel values are relatively close to the mean and exhibit low variability, despite their statistically different distributions.

One possible explanation for these variations is that different algorithms output distinct representations of mean electron impacts or electron impact counts, even after normalization. This can result in differences in observed pixel values in the aligned images and can affect pixel value distribution.

### 4.2. Performance

The time performance of the programs was evaluated by calculating the mean execution time based on 10 runs of the same simulated movie. Various movie sizes and conditions, including the number of GPUs for parallel computing, multi-threading, batch processing, and SSD storage, were considered to assess the efficiency of the algorithms in terms of processing speed.

Table 3 provides an overview of the mean performance of each algorithm based on 10 trial runs across three different Cryo-EM movie sizes. Please note that Warp was executed on a different machine than the other programs, making an absolute comparison challenging.

For the smallest movie size (4096 × 4096) with 10 frames, suitable for tomographic tilt movies, MotionCor2, FlexAlign, and Warp were the fastest, with MotionCor2 leading. In contrast, Relion MotionCor lagged, and CryoSPARC was the slowest. With 70 frames, which is common for single-particle analysis (SPA), the three fastest algorithms maintained their superiority, with Warp leading, while Relion and CryoSPARC delivered comparable performances.

As the movie size increased to 7676 × 7420, which is typical of larger movies, processing times increased significantly for all programs. In 10-frame tomographic tilt movies, the fastest algorithms’ performances resembled those of the first movie size, with MotionCor2 as the fastest. Relion MotionCor struggled the most with the larger data, while CryoSPARC was less affected but still notably slower. With 70 frames, MotionCor2’s performance declined compared to FlexAlign and Warp, being almost 10 s slower. CryoSPARC also had a notable processing time increase, while Relion MotionCor was the slowest.

For the third movie size, the increase in processing time was not as significant for all programs. In 10-frame movies, MotionCor2, FlexAlign, and Warp remained the fastest, with MotionCor2 slightly ahead. CryoSPARC ranked fourth, and Relion MotionCor was marginally slower. With 70 frames, all programs except FlexAlign and MotionCor2 experienced a substantial loss in performance, indicating difficulty in handling super-resolution-sized movies. FlexAlign consistently outperformed its peers in this scenario.

Figure 9 offers valuable insights into the scalability of alignment algorithms when utilizing GPU parallel processing. The mean processing times for different movie sizes and GPU configurations enable us to assess how efficiently these algorithms make use of parallel computing resources, which is especially important when dealing with large datasets in single-particle analysis (SPA) experiments. Three different movie sizes were tested on a machine with four GPUs.

For movie size 4096 × 4096 × 70, both FlexAlign and MotionCor2 demonstrated excellent scalability. The processing time remained nearly constant as the task was parallelized across multiple GPUs. In essence, processing one movie on one GPU took approximately the same time as processing four movies on four GPUs. However, CryoSPARC exhibited a slight decrease in performance when parallelizing the processing.

Moving to movie size 7676 × 7420 × 70, all algorithms showed a minor trade-off in performance when increasing the number of GPUs for parallel processing. This trade-off implies a small increase in processing time when using multiple GPUs compared to the ideal condition, where processing one movie on one GPU would be as fast as processing two or three movies on two or three GPUs when parallelizing. Therefore, parallelization had a minimal impact on the overall processing time for all algorithms.

For movie size 11,520 × 8184 × 70, CryoSPARC and FlexAlign continued to exhibit a slight trade-off in performance when increasing the number of GPUs, as observed in the previous sizes. However, MotionCor2 displayed a consistent pattern with no trade-off when processing more movies, indicating stable performance with parallelization. These results highlight that the scalability of alignment algorithms can vary depending on the movie size, and some algorithms may exhibit minor trade-offs in performance when parallel processing is employed.

Apart from GPU parallel computing, another approach to accelerating processing times is the use of multi-threading, a feature implemented in Relion MotionCor. Multi-threading involves allocating more CPU cores specifically for the alignment task, allowing the algorithm to leverage the computational power of multiple cores simultaneously. This can lead to improved performance and faster processing times. By optimizing the number of threads to distribute the workload efficiently across the available CPU cores, significant reductions in processing time were achieved for movies of different sizes.

For movie size 4096 × 4096 × 70, the most efficient configuration involved using one process and 36 threads, distributing the workload across 36 out of the 40 available CPU cores. This configuration reduced the processing time from 33.8 ± 5 s to 14.8 ± 0.2 s, more than halving the time required to process a movie of this size.

In the case of movie size 7676 × 7420 × 70, a similar configuration proved to be the most efficient, utilizing one process and 37 threads for workload distribution. This configuration reduced the processing time from 164 ± 7.9 s to 79.3 ± 1.1 s, again cutting the time by more than half.

Finally, for movie size 11,520 × 8184 × 70, a similar optimization was applied, reducing the processing time from 202.6 ± 22.2 s to 110.8 ± 1.2 s. This configuration involved using one process and 35 threads, decreasing the processing time by approximately a minute and a half.

Certainly, optimizing the use of around 35 threads for processing movies with 70 frames aligns well with the manufacturer’s recommendation and system capabilities. Dividing the number of movie frames (70) by the number of threads (35) results in an integer value of 2, indicating that each thread can efficiently process two frames simultaneously. This level of parallelization is the maximum achievable with the available 40 CPU cores in the system, ensuring an efficient utilization of resources.

Fine-tuning the multi-threading option in Relion, specifically for movies with 70 frames, led to a significant improvement in time performance. This optimization brought Relion MotionCor’s processing times into the same range as the faster algorithms for movie size 4096 × 4096 × 70 and significantly reduced the gap for other movie sizes. While it remained slightly slower than the fastest algorithms, this optimization made Relion MotionCor a more competitive choice in terms of processing time. These results underscore the significance of exploring alternative approaches like multi-threading to enhance the efficiency of movie alignment algorithms.

MotionCor2 offers a batch-processing feature, which can be particularly advantageous when dealing with large datasets. For this experiment, we measured the processing time of MotionCor2 for datasets comprising 20 movies of three different sizes (4096 × 4096 × 70, 7676 × 7420 × 70, and 11,520 × 8184 × 70) both using the batch processing flag and without it. Since MotionCor2 is the only algorithm that offers this option, the comparison was limited to MotionCor2 itself.

In Table 4, we can observe that batch processing significantly improves the alignment time compared to regular processing with MotionCor2.

For movies sized 4096 × 4096 × 70, using batch processing with one GPU reduced the processing time from 170 s to 100 s for the 20-movie dataset. This equates to an alignment pace of approximately 5 s per movie compared to 9 s per movie without batch processing. Further performance gains were achieved by increasing the number of GPUs to two, with a pace of 4 s per movie. However, adding more GPUs beyond this point did not yield further improvements.

As movie sizes grew to 7676 × 7420 × 70, the advantages of batch processing became even more evident. The best-case scenario, using batch processing with three GPUs, halved the pace from 25 s per movie to 12 s per movie, reducing the processing time from 740 s to 245 s. Yet, we reached a machine limit at three GPUs for this size, with no additional gains observed from further GPU additions.

For the largest movie size, 11,520 × 8184 × 70 (super-resolution size), the performance difference was even more substantial. Batch processing reduced the processing time from 38 min to 20 min, lowering the pace from 114 s per movie to 65 s per movie. Remarkably, the machine limit was reached at just one GPU for this size, indicating that the processing time was constrained by the machine’s data reading and writing capabilities, particularly when handling larger movies.

In summary, batch processing clearly accelerates the alignment process, particularly for larger movie sizes. However, it is crucial to account for machine limitations when scaling up the number of GPUs, as there may be no further performance gain beyond a certain point.

Lastly, an additional experiment was conducted involving placing data on a solid-state drive (SSD) for faster reading and writing operations. This approach aimed to further enhance the overall processing speed by leveraging the faster data transfer capabilities of an SSD.

Table 5 clearly illustrates the significant performance improvement gained by storing data on an SSD as opposed to an HDD. The alignment process demonstrated a notable speed increase, approximately 30%, which remained consistently stable. This improvement was most evident for larger movie sizes or those with more frames.

For the 4096 × 4096 movie size with 10 frames, the performance difference between the SSD and HDD was minimal, with differences in the order of tenths of seconds. However, as the number of frames increased to 70, the performance difference became more substantial, with an approximate difference of 5 s.

As the movie size increased, the performance gap between the SSD and HDD became more pronounced. For the 7676 × 7420 movie size with 70 frames, the difference was almost 10 s, highlighting the substantial advantage of SSD storage. The most significant difference was observed with the largest movie size, which is typically used for super-resolution applications. The performance gap reached approximately 20 s, underscoring the substantial impact of the data reading speed on the overall processing time.

These findings emphasize the importance of considering storage options, especially when processing data in real time or at a fast acquisition pace, and prove that utilizing an SSD for data storage can greatly enhance processing efficiency.

## 5. Discussion

Through an examination of both quality and time performance aspects, we achieved a comprehensive evaluation of the alignment programs, providing valuable insights for their practical application in cryoEM studies.

Several key points stand out based on the quality assessment of movie alignment algorithms using simulated movies (phantom movies) and real datasets (EMPIAR entries). Pattern-matching analysis on phantom movies revealed no significant differences among the algorithms in the central sections of the aligned images. However, at the edges FlexAlign consistently exhibited a cleaner pattern in both pristine and noisy images, indicating superior alignment quality. This observation was held across movies of varying sizes and frame counts. Furthermore, our findings were reinforced by results obtained using cryo-EM-based simulated movies, which replicated the same outcomes but with a cryo-EM-based signal.

The observed higher alignment quality in the central region of the movie could be attributed to factors such as the barrel deformation that gradually curves towards the edges. If this finding holds for real datasets, it could carry significant implications for cryo-EM data processing and 3D reconstruction. This improved alignment quality in the central region might lead to higher resolutions and better-quality particles, ultimately resulting in improved 3D reconstructions.

If this practical implication of the finding is confirmed, researchers could benefit from concentrating their particle selection on the central region of the image during data processing. Prioritizing particles from this area may enhance the overall alignment quality and increase the likelihood of achieving high-resolution reconstructions with superior-quality particles. However, further investigation and analysis would be necessary to pinpoint the specific reasons behind this observation.

The comparison of algorithms based on the maximum frequency of the CTF criteria on real data (EMPIAR datasets) did not reveal any consistent and significant differences when considered as a group. The ANOVA tests that showed statistical significance were not consistently reflected in both Xmipp and Gctf criteria, and these differences were not observed across all datasets. While we did not identify any statistically significant distinctions, it remains important to present these test results. Additionally, it is crucial to note that the absence of significant differences in Fourier space does not guarantee identical alignment quality in real space. Certain issues, such as misalignment along borders or at the edges, are not fully represented in the Fourier space analysis.

Analysis of the power spectrum density (PSD) trends revealed significantly greater energy dampening at high frequencies for Relion MotionCor compared to the other algorithms in all three datasets. This suggests that, regarding energy decay at higher frequencies, all algorithms except for Relion MotionCor perform similarly. Ideally, if the alignment is correct, different algorithms should not alter the PSD patterns. Therefore, when most algorithms produce similar PSDs with consistent energy decay, any algorithm that deviates introduces a bias that will propagate into subsequent 3D reconstruction steps.

When analyzing the image content in real space, we observed statistically different pixel value distributions in the aligned images when various alignment algorithms were employed. These disparities can arise from various factors, including the reference frame used by each algorithm and absolute pixel value normalization. Although the pixel value distribution does not directly impact alignment quality, it does affect the output. This complexity suggests that comparing these images is not as straightforward as simple normalization, and these differences could potentially influence subsequent processing steps, including the final reconstruction. Consequently, micrographs from different programs cannot be easily integrated into the processing pipeline. This highlights the importance of evaluating the output characteristics of each alignment algorithm when interpreting and comparing results in 3D reconstructions.

The time performance analysis of various movie alignment algorithms revealed several significant findings. Firstly, when considering individual processing times, two distinct groups of algorithms emerged based on their speed. The first group, comprising FlexAlign, MotionCor2, and Warp, exhibited significantly faster processing times compared to the second group, which included Cryosparc and Relion MotionCor. While there were no significant differences within each group, the overall performance difference between the two groups was notable, with the second group being two to four times slower.

Furthermore, when testing the algorithms with super-resolution movies, most of them experienced a significant loss in performance. FlexAlign consistently displayed the fastest performance in this scenario, being approximately 20 s faster than the closest competitor. MotionCor2 took over a minute to process such movies, while Cryosparc and Warp required over two minutes. Relion MotionCor exhibited the slowest performance, with a processing time of more than three minutes. These results indicate the challenges the algorithms face when processing larger movie sizes.

Regarding scalability, the study revealed varying degrees of scalability for different movie sizes and algorithms. Some algorithms, such as FlexAlign and MotionCor2, demonstrated excellent scalability with parallel processing. However, Cryosparc exhibited minor trade-offs in performance when parallelizing the processing. With the alternative of multi-threading, by fine-tuning the number of threads, Relion’s time performance significantly improved. Although it remained slower than the fastest algorithms for large movies, this optimization made Relion MotionCor a more competitive option in terms of processing time, highlighting the importance of exploring alternative approaches, such as multi-threading, to enhance the performance of movie alignment algorithms.

Another alternative approach was batch processing with MotionCor2. This feature demonstrated a clear advantage in accelerating the alignment process, particularly for larger movie sizes. However, it is crucial to consider machine limitations when scaling up the number of GPUs, as there may be no further increase in performance beyond a certain point. Understanding the interaction between the algorithm and machine resources is vital for optimizing the alignment process and efficiently processing large-scale cryo-electron microscopy datasets.

Additionally, the performance significantly improved when data were stored in an SSD compared to an HDD. The alignment process demonstrated notable speed enhancements and consistent stability when utilizing SSD storage. This improvement was particularly pronounced for larger movie sizes or those with more frames. The largest movie size, which is commonly used for super-resolution applications, showed the most substantial performance difference, with an approximately 20-second-per-movie advantage for SSD storage. These findings underscore the importance of considering storage options, especially in real-time processing or situations with high data acquisition rates. The utilization of SSD storage can significantly enhance processing efficiency, enabling researchers to keep pace with the demanding acquisition pace of cryo-electron microscopy data.

The execution time of a single movie alignment helps determine the required computational power for facilities. For example, if the average alignment time for typical movie sizes (let us say of size 4096 × 4096 × N) is about 13 s and the microscope produces a movie every 5 s then we would need at least 3 GPUs (3>13/5), just to cope with the movie alignment. If we also want to perform more image-processing steps along the pipeline, we may need additional GPUs. To make efficient use of the GPUs, it is important to consider implementing a queueing system to minimize idle times between movies. However, this straightforward scaling approach may not yield the expected results if the machine encounters other bottlenecks, such as insufficient CPU performance or an inadequate number of PCI-e lines to fully utilize the GPUs.

While the selection of the optimal alignment algorithm and strategy depends on multiple factors, such as the movie and dataset sizes, the trade-off between quality and performance, and the available computational capacity, we can categorize the choice into two scenarios: on-the-fly processing for cryo-EM facilities, where performance often takes precedence over quality, and structural processing, where achieving the highest quality results outweighs performance considerations.

For both scenarios, we recommend using SSD storage; however, it should be noted that for large cryo-EM projects SSD capacity may quickly become insufficient due to its typically smaller size, which, of course, depends on each specific case. Additionally, we recommend GPU computing with parallelization limited to 2–3 GPUs, as it is likely to reach the machine’s limit regardless of the number of GPUs used.

Regarding algorithm selection, in the context of real-time processing, our recommendation leans towards MotionCor2 with batch processing due to its faster performance. However, for more complex structural processing, if you must make a single choice, we suggest FlexAlign. It appears to deliver superior alignment across all movie regions.

When it comes to complex structural processing, it is worth noting that movie alignment algorithms exhibit diverse error patterns. Some, like Relion MotionCor, Warp, or FlexAlign, demonstrate independent errors, such as energy dampening at high frequencies, specific issues with certain datasets, or sensitivity to high drift, respectively. In such cases, the mistakes made by one algorithm are not necessarily related to the mistakes made by another and experimenting with different algorithms can help uncover these unique issues.

However, there are situations where multiple algorithms make correlated mistakes. For example, most algorithms tested in this article struggled with aligning the movie edges of the phantom movies. When several algorithms encounter the same underlying issue, comparing their outcomes might not provide a solution if the same underlying issue is affecting all of them. However, exceptions can occur, as observed with FlexAlign’s superior alignment at the edges.

For both contexts, consensus algorithms hold value but face challenges in merging results due to varying pixel value distributions in aligned images. Instead, practicality lies in comparing results and evaluating the alignment agreement. Any disparities may suggest errors or a superior performance by one algorithm. In practice, relying on consensus algorithms should be based on empirical testing. It is essential to ensure that the results from multiple algorithms align with your findings, especially in cryo-electron microscopy, where the ground truth is often unknown.

Finally, we believe that the algorithms discussed in this article have undergone extensive development, taking years to refine and correct errors while enhancing their capabilities. These algorithms are among the most well known and widely used worldwide, and they are continually evolving. Therefore, we believe that investing substantial time in developing entirely new movie alignment algorithms may not be the most efficient approach. Instead, contributing to the ongoing improvement of existing methods would be a more valuable way to advance the field.

## 6. Conclusions

Overall, our study provides critical insights into the strengths and weaknesses of various movie alignment algorithms in cryoEM, contributing to the understanding and selection of suitable algorithms and the most efficient approach for cryo-electron microscopy studies. Additionally, we recommend considering alternative approaches, such as multi-threading and batch processing, to optimize alignment performance and improve efficiency in large-scale data processing. Moreover, attention to machine resources and storage options is crucial for successful cryoEM data processing, ensuring researchers can keep up with the demanding pace of data acquisition.

As future research lines, it would be interesting to extend the study and investigate how different file formats may affect the performance of movie alignment algorithms. Different movie file formats may have distinct compression methods, data organization, and metadata, which can influence the processing time.

We also provide all scripts and data used for this study with the hope that they can be used as a reference point for comparison of future versions of movie alignment software. As we do not consider ourselves experts on the behavior of all tested programs, their authors can, for example, use our data to explain different options and provide presets for different movie sizes, optimizing both the quality of the alignment and the execution time.

## Figures and Tables

**Figure 1 micromachines-14-01835-f001:**
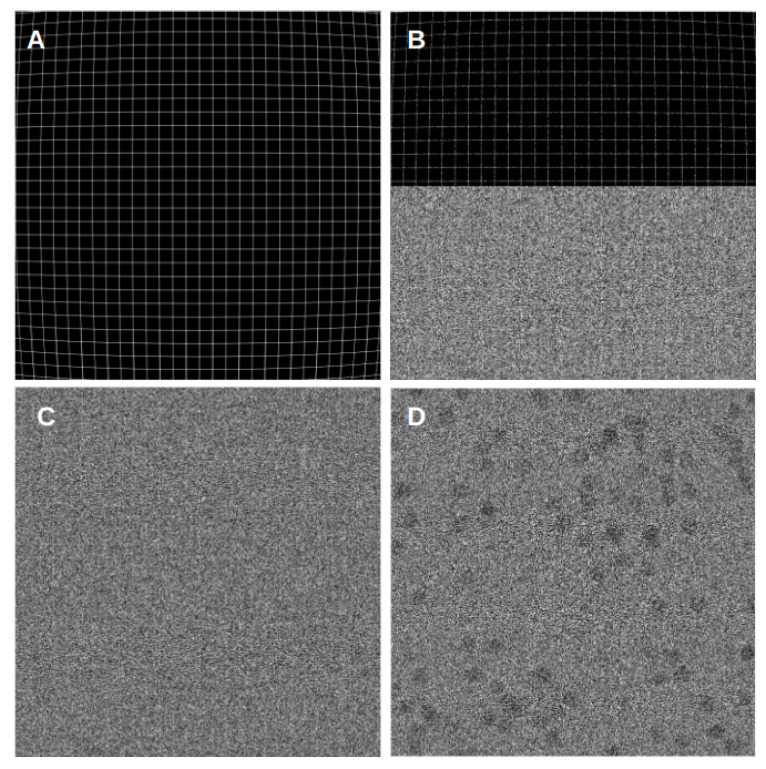
Phantom movie generation process, noiseless frame, effects of applied noise and dose models, and a cryo-EM-based simulated movie frame. (**A**) corresponds to a frame of a noiseless movie with the aforementioned barrel deformation. (**B**) represents the same frame, with the upper half showing the impact of the corresponding applied dose and the lower half displaying the effect of the noise model simulating ice. (**C**) depicts the final generated movie frame with the applied dose and the corresponding noise model. (**D**) shows a cryo-EM-based simulated movie frame with circular projections, dose, and noise model. For representation, this frame was enhanced in contrast and Gaussian blurred to make the simulated particles visible.

**Figure 2 micromachines-14-01835-f002:**
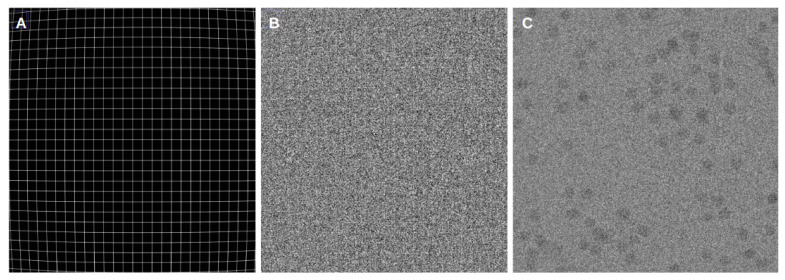
Examples of different aligned phantom movies: (**A**) shows frames aligned from a noiseless movie using the FlexAlign program. (**B**) depicts frames aligned from the same movie, but this one includes an additional noise model, also aligned using the FlexAlign program. (**C**) illustrates frames aligned with the FlexAlign program from a cryo-EM-based simulated movie with disc projections.

**Figure 3 micromachines-14-01835-f003:**
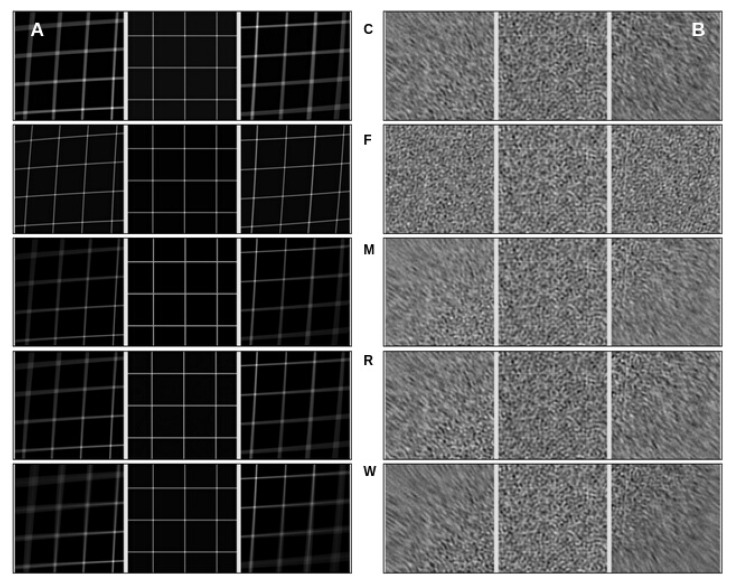
Pattern matching. The visual analysis of alignment algorithm results was conducted on a phantom movie with dimensions of 4096 × 4096 × 70. Each panel (**A**,**B**) in the figure displays a set of images, where each row represents a triplet of images extracted from the top-left, center, and bottom-right parts of the aligned movies. (**A**) The left panel showcases the alignment of noiseless movies, whose frames contain deformed and shifted mesh patterns. Each row corresponds to a run of a specific algorithm, arranged alphabetically from CryoSPARC (C) to Warp (W) software, showing the first letter of their name in the middle of the row. (**B**) The right panel features the alignment of movies of the same dimensions and mesh pattern but with added noise.

**Figure 4 micromachines-14-01835-f004:**
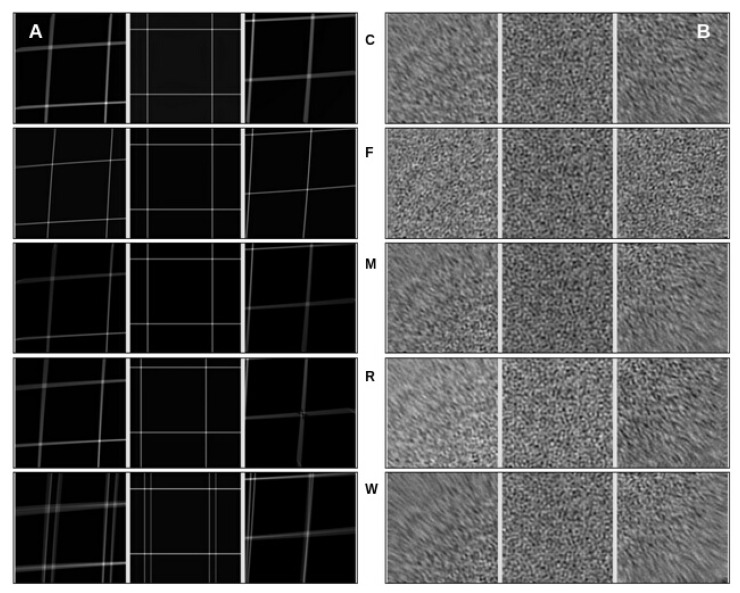
The effect of a 60-pixel total shift on alignment algorithms was analyzed using a phantom movie with dimensions of 4096 × 4096 × 70. Each panel (**A**,**B**) in the figure presents a set of images, where each row represents a triplet of images extracted from the top-left, center, and bottom-right parts of the aligned movies. (**A**) The left panel displays the pristine (noiseless) movies, where frames contain deformed and shifted mesh patterns. Each row corresponds to a run of a specific algorithm, arranged alphabetically from CryoSPARC (C) to Warp (W) software, showing the first letter of their name in the middle of the row. (**B**) The right panel features movies of the same dimensions and mesh pattern but with the addition of noise.

**Figure 5 micromachines-14-01835-f005:**
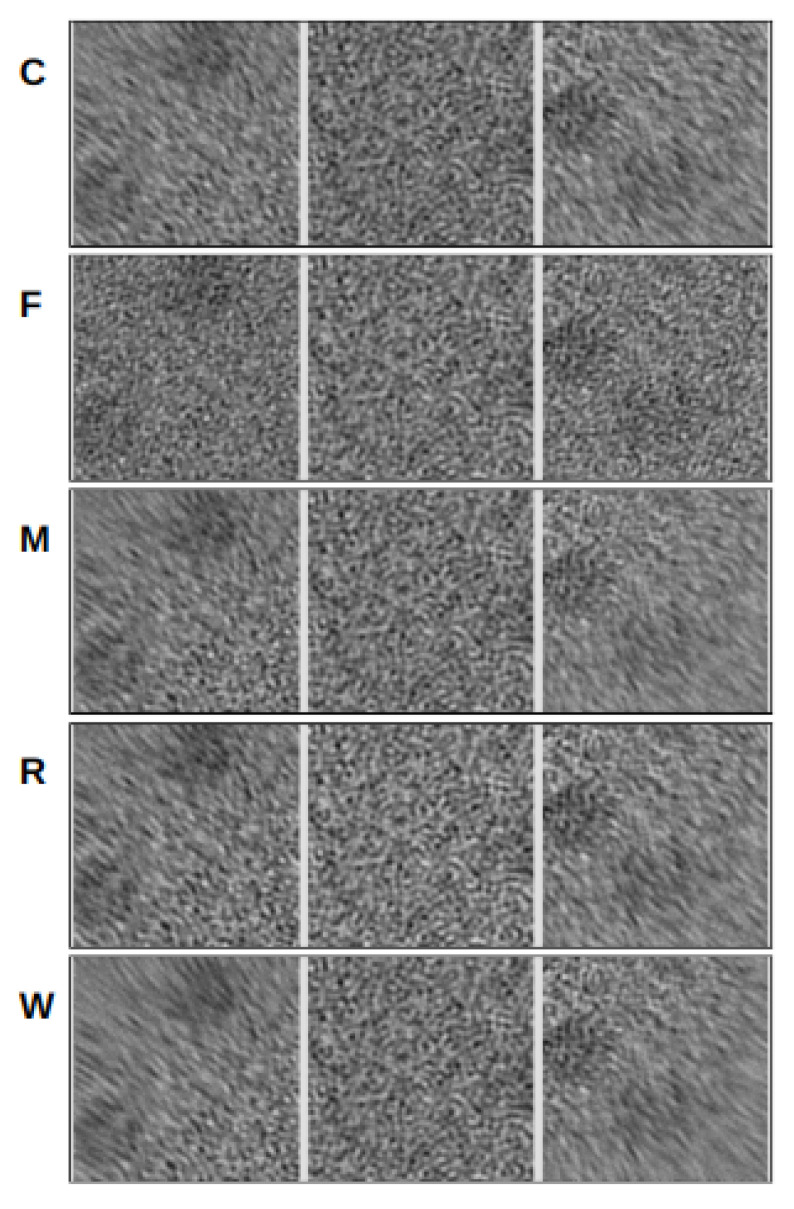
Alignment of cryo-EM-based simulated movies. Visual analysis of alignment algorithm results was conducted on a phantom movie with dimensions of 4096 × 4096 × 70. The figure displays a set of images, with each row representing a triplet of images extracted from the top-left, center, and bottom-right parts of the aligned movies. This movie contains circular projections to simulate particles, dose, noise, and deformations to fully replicate cryo-EM conditions. Each row corresponds to a specific algorithm run, arranged alphabetically from CryoSPARC (C) to Warp (W) software, with the first letter of their name shown in the middle of the row.

**Figure 6 micromachines-14-01835-f006:**
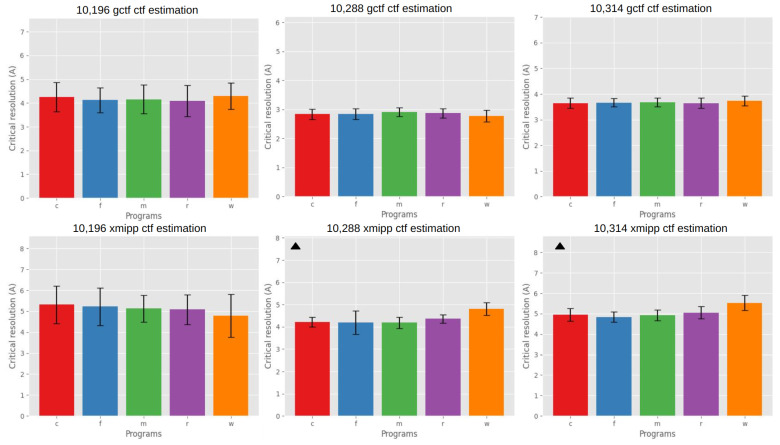
Resolution based on CTF Criteria. The bar plots in the first row depict the mean CTF criteria obtained with the Gctf program for EMPIAR entries 10,196, 10,314, and 10,288. Each bar corresponds to the specific alignment algorithm result, arranged alphabetically: CryoSPARC (C), FlexAlign (F), MotionCor2 (M), Relion MotionCor (R), and Warp (W). Similarly, the second row contains bar plots for each entry, but, in this case, the mean critical resolution is estimated using the Xmipp program. A triangle symbol indicates a significant difference between means (ANOVA, confidence value 0.05).

**Figure 7 micromachines-14-01835-f007:**
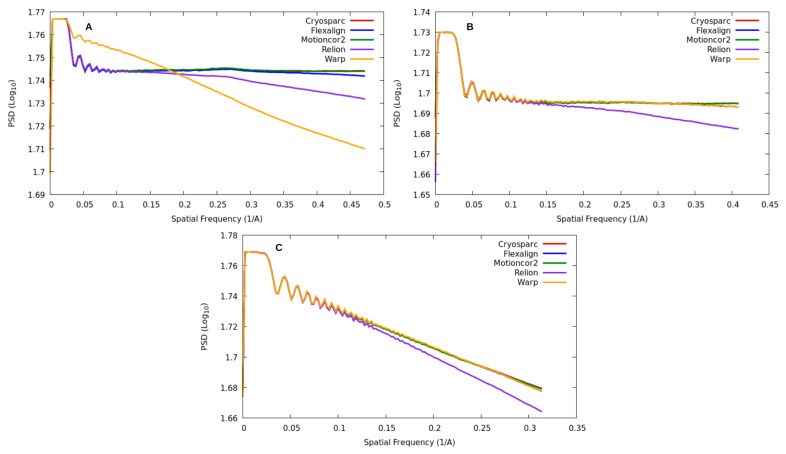
Power spectrum density (PSD) trends for all the programs are depicted in the figures. Each figure corresponds to one of the EMPIAR entries and illustrates the PSD plots of a single micrograph aligned using different algorithms. It is worth noting that we observed a consistent trend for each program across all the micrographs in our datasets. To illustrate this behavior, we randomly selected one image per entry to represent the overall trend. (**A**) displays the PSD plots, presented in a logarithmic scale, of a sample image from EMPIAR 10,196 utilizing various algorithms. (**B**) showcases the PSD plots of a sample image from EMPIAR 10,288. (**C**) exhibits the PSD plots of a sample image from EMPIAR 10,314.

**Figure 8 micromachines-14-01835-f008:**
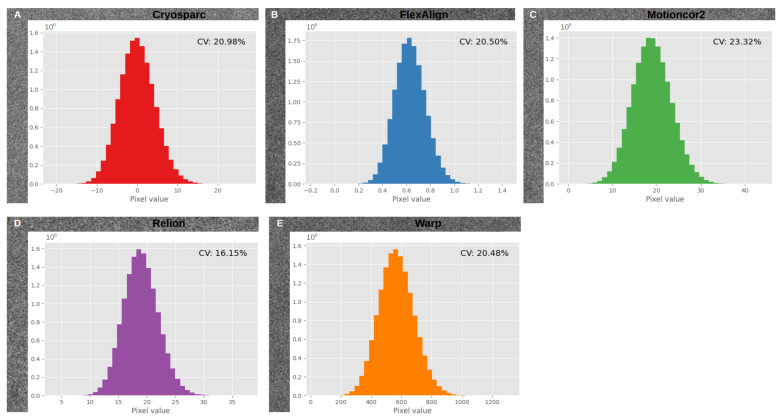
Aligned movies’ pixel values histograms are presented for the same aligned movie from the EMPIAR entry 10,288: (**A**) Cryosparc; (**B**) FlexAlign; (**C**) Motioncor2; (**D**) Relion MotionCor; (**E**) Warp. CV stands for coefficients of variation (standard deviation as a percentage of the mean). For representation, we removed the outliers by means of the interquartile range (IQR of 80%) method.

**Figure 9 micromachines-14-01835-f009:**
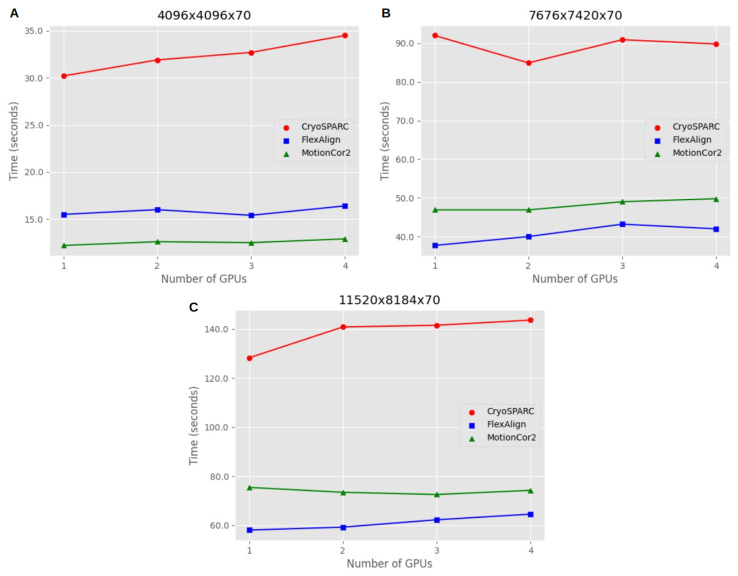
Scalability of parallel GPU processing. The plots represent the mean processing time in seconds (*y*-axis) required to process a single movie on one GPU. The *x*-axis represents the number of GPUs, which is increased in parallel with the number of movies to process. This figure demonstrates the scalability of the algorithm with GPU parallel processing. The scalability analysis was performed on three different movie sizes commonly encountered in single-particle analysis (SPA) experiments. These movie sizes are as follows: (**A**) corresponds to the 4096 × 4096 × 70 experiment, which represents a movie size typically observed in lower-resolution SPA experiments; (**B**) represents the 7676 × 7420 × 70 experiment; (**C**) corresponds to the 11,520 × 8184 × 70 experiment, which is a movie size commonly used in super-resolution acquisitions. To ensure reliable results, each algorithm was executed 10 times per movie size, thereby avoiding unstable runs and obtaining more accurate measurements.

**Table 1 micromachines-14-01835-t001:** Comparison of various movie alignment programs.

Program	HW	Method + Interpolation
CryoSPARC	GPU	Proprietary code
FlexAlign	GPU	CC + cubic B-spline in space and time
MotionCor2	GPU	CC + quadratic (space), cubic (time) polynomials
Relion MotionCor	CPU	CC + quadratic (space), cubic (time) polynomials
Warp	GPU	CC + cubic B-spline in space and time

**Table 2 micromachines-14-01835-t002:** CTF Resolution limit (Å) comparison. The following table presents the means and standard deviations of the CTF criteria for the CTF estimation using two different methods, Gctf and Xmipp. The data in the table correspond to 30 image samples per EMPIAR entry, divided into three datasets: 10,196, 10,288, and 10,314.

EMPIAR Entry	Program	Xmipp	Gctf
	CryoSPARC	5.3 ± 0.9	4.3 ± 0.6
10,196	FlexAlign	5.2 ± 0.9	4.1 ± 0.5
	MotionCor2	5.1 ± 0.7	4.2 ± 0.6
	Relion MotionCor	5.1 ± 0.7	4.1 ± 0.7
	Warp	4.8 ± 1	4.3 ± 0.6
	CryoSPARC	4.2 ± 0.2	2.8 ± 0.2
10,288	FlexAlign	4.2 ± 0.5	2.8 ± 0.2
	MotionCor2	4.2 ± 0.3	2.9 ± 0.2
	Relion MotionCor	4.4 ± 0.2	2.9 ± 0.2
	Warp	4.8 ± 0.2	2.8 ± 0.2
	CryoSPARC	5.0 ± 0.3	3.6 ± 0.2
10,314	FlexAlign	4.8 ± 0.3	3.7 ± 0.2
	MotionCor2	4.9 ± 0.3	3.7 ± 0.2
	Relion MotionCor	5.1 ± 0.3	3.6 ± 0.2
	Warp	5.5 ± 0.4	3.7 ± 0.2

**Table 3 micromachines-14-01835-t003:** This table provides the time performance comparison of different movie alignment software for CryoEM. It presents the mean and standard deviation of the time in seconds required to align movies of various sizes (4096 × 4096, 7676 × 7420, and 11,520 × 8184). The data in the table represent the results obtained from performing the alignment process 10 times. The movie sizes are based on three sizes commonly encountered in cryo-electron microscopy tomography and single-particle analysis (SPA), each with a different number of frames, 10 and 70 frames, respectively.

Movie Size	Program	10 Frames	70 Frames
	CryoSPARC	15.9 ± 1.1	30.2 ± 3.4
	FlexAlign	3.8 ± 0.1	15.5 ± 0.9
4096 × 4096	MotionCor2	3.2 ± 0.2	12.2 ± 1.0
	Relion MotionCor	7.9 ± 0.7	33.8 ± 5.0
	Warp	3.9 ± 0.5	11.6 ± 1.5
	CryoSPARC	27.8 ± 2.4	92 ± 6.7
	FlexAlign	9.8 ± 0.5	37.7 ± 1.1
7676 × 7420	MotionCor2	7.8 ± 0.3	46.9 ± 1.9
	Relion MotionCor	37.1 ± 2.3	164 ± 7.8
	Warp	10.7 ± 1.5	36.7 ± 2.9
	CryoSPARC	42.2 ± 2.6	128.3 ± 8.8
	FlexAlign	14.1 ± 0.8	58.1 ± 2.1
11,520 × 8184	MotionCor2	11.9 ± 0.4	75.5 ± 3.8
	Relion MotionCor	46.4 ± 3.1	202.6 ± 22.2
	Warp	16.9 ± 2.1	130.9 ± 4.0

**Table 4 micromachines-14-01835-t004:** This table compares the time performance in seconds for processing data with and without the MotionCor2 batch processing option. The size of the batch is managed internally by the program; it only requires the directory where the movies are located. The mean and standard deviation of the time required to align an entire dataset of movies with different sizes (4096 × 4096 × 70, 7676 × 7420 × 70, and 11,520 × 8184 × 70) are presented. The data in the table represent the results obtained from performing this process five times.

Movie Size	Regular Processing	Batch Processing	GPU
4096 × 4096 × 70	170.6 ± 3.5	100.7 ± 7	1
		82.1 ± 1.2	2
		84.1 ± 5.1	3
		89.4 ± 2.7	4
7676 × 7420 × 70	740.5 ± 38.8	509.7 ± 71.9	1
		279.6 ± 2.2	2
		245.7 ± 9.4	3
		258.2 ± 8.8	4
11,520 × 8184 × 70	2288.7 ± 127.9	1309.4 ± 20.2	1
		1442.7 ± 137	2
		1317.6 ± 29.2	3
		1331.8 ± 18.6	4

**Table 5 micromachines-14-01835-t005:** This table presents a time performance comparison in seconds between processing data stored in HDD (hard disk drive) and SSD (solid-state drive) storage using the FlexAlign movie alignment program. The mean and standard deviation of the time required to align movies of various sizes are provided, specifically for movies of 4096 × 4096, 7676 × 7420, and 11,520 × 8184. The data in the table represent the results obtained from performing the alignment process 10 times. The movie sizes are based on three standard sizes commonly encountered in cryo-electron microscopy tomography and single-particle analysis (SPA), with different numbers of frames (10 and 70 frames, respectively).

Movie Size	Frames	HDD	SDD
4096 × 4096	10	3.8 ± 0.1	3.2 ± 0.1
	70	15.5 ± 0.9	10.6 ± 0.3
7676 × 7420	10	9.8 ± 0.5	7.4 ± 0.4
	70	37.7 ± 1.1	26.8 ± 1.2
11,520 × 8184	10	14.1 ± 0.8	11.4 ± 0.8
	70	58.1 ± 2.1	40.2 ± 1.8

## Data Availability

This paper analyzes existing, publicly available data in EMPIAR. These accession numbers for the datasets are reported in the paper (10,196, 10,288, and 10,314). The generated phantom movies have been deposited at EMPIAR under accession code [PENDING FOR ACCEPTANCE]. The code has been deposited at Zenodo and is publicly available (accessed on 1 August 2023) before accepted date of publication (19 September 2023). The DOI is https://doi.org/10.5281/zenodo.8186837. The software is also freely available from https://github.com/danielmarchan3/ComparisonMovieAlignment-cryoEM.git, accessed on 1 August 2023. Any additional information required to reanalyze the data reported in this paper is available from the lead contact upon request.

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
