# Peer review of "Performance and Quality Comparison of Movie Alignment Software for Cryogenic Electron Microscopy"

_micromachines, 2023, doi:10.3390/mi14101835_

Round 1

Reviewer 1 Report

The paper aims to assess the performance of various movie alignment programs used for cryo-electron microscopy data processing. As there is a “zoo” of available programs, it is important to decide which program to use, and in which mode. In that sense, the paper represents a very important potential continuation.

However, I have two major concerns regarding the work. First, the authors fail to convince that they use an appropriate methodology to evaluate the performance of the different programs. Second, even if we accept the proposed methodology, there seems to be missing a “bottom line” for the research – what is the recommendation? Which program should be used and in which cases? Should we bother developing better movie alignment algorithms, or do the existing algorithms good enough? Do the algorithms make correlated or independent mistakes, so that there is a point in devising a consensus algorithm to combine their outcomes? What are the limitations of the existing algorithms, for examples, in terms of SNR?

Regarding their evaluation methodology, the authors try to evaluate the quality and speed of the different programs. While the evaluation of speed is adequate, the evaluation of their quality is not. Specifically, the methodology use to evaluate the algorithms does not convince that we actually learn anything from the presented results.

Methodological issues:

1.      The simulated phantom used to evaluate the different algorithms is very different from how real data looks. What can we learn about the performance of the different algorithms from data which is very different from their typical input? For example, the simulated phantom is very periodic, and contains essentially only high frequencies. Why no using simulated projections of a known density?

2.      Why is energy decay at high frequencies relevant to alignment accuracy? Where was this shown?

3.      Comparing the achievable resolution using the CTF curves does seem to teach us anything about differences between the algorithms.

4.      I did not understand what information we gain from the histograms in Figure 4. A simple normalization of the image will change the histogram without changing its information content. Moreover, what do these histograms have to do with the quality of the alignment?

5.      In Figure 5, the right panel (a) is the ground truth or the output of some alignment algorithm (as the caption suggests)? If it is the output of an alignment algorithm, then how was it generated? Similarly, how was panel (b) generated?

6.      Figure 6: It is unclear how the left panel was generated. Is it by applying the algorithms to clean data? What do we learn visually from panel B?

7.      On line 363 it is said that Figure 6 “allows for the evaluation of … the impact of noise on the alignment quality”. How do we assess the impact of noise from Figure 6?

8.      The claim “This enhanced alignment quality in the center region may result in higher resolution and better quality particles, ultimately leading to improved 3D reconstructions” (line 573) which is potentially very important, was not validated. For example, is this claim apparent in the CTF of particles from these regions?

Minor issues:

1.      Figure 2: There is no legend (nor text) to explain the letter coding on the x-axis

2.      Figure 2: I did not understand what are the lines on the two right panels at the bottom row. Why there are two lines in each of the panels?

3.      Line 330: What does it mean “frequency distribution of pixel values”?

4.      On line 336 it is said that “there were no significant changes observed in the different distribution of their data”. In the following paragraph, it is say that “One possible explanation for these variations…” Don’t the two paragraph contradict each other? Why to explain the variability if there is no variability?

5.      On line 344, the sentence “The variations in pixel values highlight the importance of understanding and considering the specific output characteristics of each alignment algorithm when interpreting and comparing the results” is unclear. How does it highlight the importance? What are “output characteristics”?

6.      On line 391 it is said that “FlexAlign, in particular, was sensitive to these changes”. How is this line consistent with line 371 which says “FlexAlign stood out by achieving a cleaner pattern”?

7.      Section 4.2 describes in many words things that are easily seen from Table 3. Wouldn’t it be more beneficial to describe just the conclusions from the experiment?

8.      What do the authors mean by “a small trade-off in performance” (for example, in line 455)?

9.      The authors claim on line 545 that their study results in “providing valuable insights for their practical use”. However, I could not find such insights.

10.   The paper requires professional English editing. While the paper is readable and accessible, there are various minor issues that should be fixed.

N/A

Author Response

We appreciate the reviewer's time and recommendations. With your help, we have created a more complete and helpful manuscript. As for every recommendation, we have addressed them in the attached document.

Reviewer 2 Report

In the submitted manuscript, the authors analyzed and compared the quality and performance of some of the most commonly used software packages for Cryo-EM movie alignment. They have tested the alignment quality and performance on both simulated datasets and real data and compared the alignment precision, power spectrum density and performance scaling of each algorithm. This analysis has provided some critical insights into the strengths and weaknesses of the different programs and could be used as a guide for choosing the right one for different Cryo-EM studies. I would recommend the manuscript for publication in its present form.

One minor comment: The acronym „SW“ on line 68 (I assume SW stands for Software) hasn’t been previously defined.  

Author Response

We appreciate the reviewer's time and recommendations. With your help, we have created a more complete and helpful manuscript. As your  recommendation, it was addressed in this version. We are glad that you like it.

Reviewer 3 Report

Cryo-EM plays a crucial role in structural biology, and it is important to process the so-called movie data produced by the microscopes in real-time. This manuscript compared the quality and performance of the most commonly used software for movie alignment. I have the following suggestions and comments:

1. Parameters used in the experiment should be discussed. For example, how are the values of a1, a2, b1, and b2 in Eq(1) determined? Authors should give reasons or references.

2. In line 143, how are the noisy and noiseless movies generated? Is there any software to generate them and how to add the noise?

3. It will be better to report the total average results in Table 2.

4. 3D reconstruction is an important step after the cryo-EM data processing. Authors should discuss them in Introduction or Discussion. The following related works should be include in the discussion: DeepTracer( PNAS, 2021, 118, e2017525118), DEMO-EM(Nature Comput. Sci., 2(3): 265-275), and RosettaES (Nat. Methods 2017, 14, 797-800).

No.

Author Response

(The authors gave the same response as above.)

Round 2

Reviewer 1 Report

None.

None.